# Determinants of QTc Interval Prolongation in Patients with Hypopituitarism and Other Pituitary Disorders

**DOI:** 10.3390/biomedicines13112676

**Published:** 2025-10-31

**Authors:** Valentina Gasco, Daniela Cuboni, Sergio Siclari, Francesca Mocellini, Michela Sibilla, Silvia Grottoli, Ezio Ghigo, Mauro Maccario

**Affiliations:** Department of Medical Science, Division of Endocrinology, Diabetes and Metabolism, University of Turin, 10126 Turin, Italy; daniela.cuboni@unito.it (D.C.); sergio.siclari@edu.unito.it (S.S.); francesca.mocellini@unito.it (F.M.); michela.sibilla@unito.it (M.S.); silvia.grottoli@unito.it (S.G.); ezio.ghigo@unito.it (E.G.); mauro.maccario@unito.it (M.M.)

**Keywords:** hypopituitarism, hypocortisolism, hypothyroidism, growth hormone deficiency, diabetes insipidus, QT interval, acquired long QT syndrome, electrolyte imbalance, cardiac arrhythmia

## Abstract

**Background**: Long QT syndrome (LQTS) is characterized by delayed myocardial repolarization, predisposing to malignant arrhythmias such as torsades de pointes, ventricular fibrillation, and cardiac arrest. Recent reports suggest that acquired LQTS (aLQTS) may represent an early manifestation of hypopituitarism, potentially contributing to its increased cardiovascular mortality, although evidence remains limited to 16 published case reports. **Objective**: The objective was to investigate the relationship between hypopituitarism and corrected QT (QTc) interval. **Methods**: We retrospectively analyzed data from 185 patients (121 males) with hypothalamic–pituitary disorders who underwent a 12-lead electrocardiogram between April 2023 and September 2024. Clinical characteristics, hormone replacement therapy, and same-day laboratory parameters (electrolytes, fT3, fT4, IGF-I, testosterone) were recorded. QTc was calculated using Bazett’s formula. Multivariate logistic regression identified predictors of QTc prolongation. **Results**: Age (OR 1.07–1.09, *p* = 0.02) was a significant predictor in 5 of 8 models. The presence of expansive lesions other than pituitary adenomas, craniopharyngiomas, and Rathke’s cleft cysts was also associated with QTc prolongation (OR 8.35–17.73, *p* < 0.05 and *p* = 0.03). Potassium (OR 0.14–0.17, *p* = 0.09) and albumin-corrected calcium levels (OR 0.0003, *p* = 0.06) showed consistent, though borderline, associations. **Conclusions**: Age and the presence of expansive lesions other than pituitary adenomas, craniopharyngiomas, and Rathke’s cleft cysts are the main predictors of QTc duration in patients with hypothalamic–pituitary disease. Electrolyte imbalances—particularly low potassium and albumin-corrected calcium—may further contribute. The influence of specific pituitary deficiencies remains uncertain, likely due to adequate replacement therapy in most patients.

## 1. Introduction

Hypopituitarism is characterized by a complete or partial deficiency in the secretion of one or more pituitary hormones and may result from primary pituitary disease or hypothalamic disorders [1,2,3,4,5]. When all pituitary hormones are deficient, the condition is termed panhypopituitarism. This rare disorder can occur at any age and often progresses insidiously over time, although its onset may be acute [1,2,3,4,5]. Clinical manifestations are highly variable and depend on the underlying etiology, the specific hormonal deficits, their severity, the rate of onset, and the patient’s age [1,2,3,4,5]. Symptoms are typically chronic and nonspecific; however, in certain cases, hypopituitarism may present acutely and constitute a life-threatening emergency [1,2,3,4,5].

Patients with hypopituitarism have been reported to be at increased risk of cardiovascular morbidity and mortality, particularly among women and individuals diagnosed at a young age [6,7,8,9,10,11]. This elevated risk appears to be at least partly attributable to suboptimal management of hormone replacement therapies, whether due to omission (as in cases of gonadotropin or growth hormone deficiency) or the use of supra- or subphysiological doses (as in adrenal or thyroid hormone replacement) [10,11,12,13,14,15]. In addition, the underlying etiology of the hypothalamic–pituitary disorder may independently contribute to the higher cardiovascular morbidity and mortality observed in these patients compared with the general population [7,8,9]. Long QT syndrome (LQTS) is a cardiac disorder characterized by delayed repolarization of myocardial cells, which can lead to malignant arrhythmias such as torsade de pointes, ventricular fibrillation, or even sudden cardiac arrest [16,17]. The QT interval represents the time required for ventricular depolarization and repolarization. QT prolongation reflects an extended action potential duration in ventricular cardiomyocytes and may result from either a reduction in outward repolarizing currents—typically due to impaired potassium channel function—or an increase in inward depolarizing currents, linked to enhanced sodium or calcium channel activity [16,17,18,19,20,21,22,23]. Although a corrected QT (QTc) interval ≥ 480 milliseconds (ms) is commonly used as the threshold for defining QT prolongation [16], no lower limit has been established below which QTc prolongation can be considered free of proarrhythmic risk [17]; a QTc interval ≥ 450 ms in males and ≥460 ms in females is usually regarded as borderline [16]. Congenital LQTS (cLQTS) arises from mutations in genes encoding specific ion channels or their regulatory proteins [18,19,20,21,22,23], whereas acquired LQTS (aLQTS) results from dysfunction of the same ion channels due to environmental factors—such as medications, electrolyte disturbances, or concomitant heart disease—in the absence of pathogenic genetic mutations [17,24,25].

QTc prolongation is observed in hypothalamic–pituitary endocrine disorders and is particularly well-documented in hypersecretory conditions. Prolonged QTc has been consistently documented in patients with Cushing’s syndrome (CS) compared with healthy controls [26]. In CS, hypercortisolism and hypokalemia represent independent risk factors for left ventricular remodeling and dysfunction [27,28], both conditions associated with increased QT duration [27,28,29,30,31]. In the cohort reported by Pecori Giraldi et al. [32], QTc prolongation was observed in 26% of male patients with CS and was significantly associated with hypokalemia and reduced testosterone levels, whereas no cases were detected among female patients or healthy controls.

Cardiac rhythm abnormalities and QTc prolongation are common and clinically relevant in acromegaly, potentially affecting quality of life and survival [33,34,35,36]. These abnormalities are thought to result primarily from myocardial interstitial fibrosis induced by growth hormone (GH)–insulin-like growth factor I (IGF-I) excess. Ventricular arrhythmias are more frequent and severe than in controls, correlating with disease duration and left ventricular mass but not with circulating GH levels [34,35,37,38,39].

Although the direct relationship between hyperprolactinemia and QT prolongation requires further study, evidence suggests that elevated prolactin levels, particularly when drug-induced, may contribute to QT interval prolongation [40]. Experimental data also indicate that prolactin can prolong ventricular repolarization in genetically predisposed individuals, thereby increasing susceptibility to arrhythmic events [41].

In recent years, several case reports have suggested that QTc interval prolongation may represent an early clinical manifestation of hypothalamic–pituitary hypofunction and may, at least in part, contribute to the increased cardiovascular mortality observed in patients with hypopituitarism. However, the mechanisms underlying this abnormality in hypopituitary patients have not been fully elucidated. Indeed, the current literature on the potential association between hormonal alterations in hypopituitarism and aLQTS remains limited; to date, only 16 case reports have been published [42,43,44,45,46,47,48,49,50,51,52,53,54,55,56,57], describing both male and female patients who presented with QTc prolongation at diagnosis. These cases varied in terms of the etiology and severity of hypopituitarism (i.e., the number and type of affected hormonal axes) as well as the clinical severity at presentation. In some instances, concomitant electrolyte disturbances at diagnosis could independently account for the QTc prolongation [43,49,50,53,54]. Additionally, other predisposing clinical conditions or the use of QT-prolonging drugs were reported in some patients [43,44,46,51,55,56,57].

Although the precise mechanisms underlying the development of aLQTS in patients with hypopituitarism are not fully understood, thyroid and adrenal hormone deficiencies are currently considered the most likely pathophysiological contributors [42,43,44,45]. Regarding sex hormones, it is important to note that QT interval duration is similar in boys and girls during childhood but becomes shorter in males after puberty [17]. Experimental studies suggest that testosterone is a key mediator of this difference. Indeed, in male patients with either primary or secondary hypogonadism and aLQTS, QTc prolongation has been shown to normalize with testosterone replacement therapy [42,43,58]. Moreover, adult females with cLQTS types 1 and 2 are at greater risk of polymorphic ventricular tachycardia and sudden cardiac death, suggesting a possible pro-arrhythmogenic role of female sex hormones in modulating risk [59]. Based on these observations, it has been hypothesized that estrogen deficiency might exert a protective effect against QT prolongation. Finally, there is currently no evidence supporting a direct role of growth hormone deficiency (GHD) in the development of aLQTS in patients with hypopituitarism.

This study aimed to investigate the relationship between hormonal deficiencies and QTc interval duration in patients with hypothalamic–pituitary disorders. Specifically, we assessed whether hypopituitarism per se is associated with QTc prolongation. To this end, we conducted a retrospective analysis of a cohort of patients with varying degrees of hypopituitarism, with or without hormone replacement therapy, followed at a tertiary referral center for the management of pituitary disorders.

## 2. Materials and Methods

A retrospective analysis was performed on data from patients with hypothalamic–pituitary disorders who underwent a standard 12-lead electrocardiogram (ECG) at the Endocrinology, Diabetes, and Metabolism Unit of the “Città della Salute e della Scienza” Hospital, Turin, Italy, between 1 April 2023, and 30 September 2024.

All patients with a documented history of hypothalamic–pituitary disease were considered, irrespective of the presence or absence of hypopituitarism. To specifically evaluate the impact of hypopituitarism on QT interval duration, individuals with pituitary adenomas associated with active hormonal hypersecretion (Cushing’s disease, acromegaly, hyperthyroidism, and hyperprolactinemia) were excluded. Patients receiving medications with potential QT-prolonging effects, those with a family history of sudden cardiac death, and those with a personal history of cardiovascular events were also excluded from the analysis. The list of medications potentially associated with QT prolongation was obtained from the website of the Arizona Center for Education and Research on Therapeutics (https://www.azcert.org/, accessed on 30 August 2025).

For each patient, in addition to ECG data used for QTc interval assessment, the following information was collected: sex, age, underlying hypothalamic–pituitary pathology, presence and type of pituitary hormone deficiencies, and any ongoing hormone replacement therapies. When available on the same day as the ECG, serum concentrations of sodium, potassium, calcium, albumin, free triiodothyronine (fT3), free thyroxine (fT4), insulin-like growth factor I (IGF-I), and testosterone (in males) were analyzed.

ECGs were recorded using a Schiller Cardiovit AT-102 G2 electrocardiograph (Schiller AG, Baar, Switzerland). The QTc interval was calculated using Bazett’s formula: QTc (ms) = QT (ms)/√(RR interval/60). QTc prolongation was defined as ≥450 ms in males and ≥460 ms in females [16].

### 2.1. Biochemical and Hormonal Assessments

All biochemical analyses were performed using standard methods at the Central Laboratory of the hospital. Derived parameters included albumin-corrected calcium, calculated as follows: albumin-corrected calcium (mmol/L) = 0.02 × (40 − serum albumin [g/L]) + serum calcium (mmol/L), and IGF-I standard deviation scores (SDS), calculated according to the formula {(IGF-I/M)^L^ − 1}/(L × S), using normative data from Bidlingmaier et al. based on more than 15,000 healthy subjects aged 0–94 years [60].

All blood sampling and instrumental evaluations were performed in the morning after an overnight fast of at least 10 h, with procedures initiated between 8:00 and 9:00 a.m.

All samples from each individual were analyzed together.

All participants provided written informed consent. The study was approved by the local Ethics Committee (protocol code 0040828).

### 2.2. Statistical Analysis

Continuous variables were expressed as mean ± standard deviation (SD) for normally distributed data and as median with interquartile range (IQR) for non-normally distributed data, while categorical variables were presented as percentages. Normality was assessed using the D’Agostino–Pearson test.

Comparisons between independent groups were performed using Student’s *t*-test for normally distributed variables and the Mann–Whitney U test for non-normally distributed variables. Correlations were evaluated using Pearson’s correlation coefficient for normally distributed data and Spearman’s rank correlation coefficient for non-normally distributed data. The Chi-square test or Fisher’s exact test was used to assess significant differences between categorical variables.

The association between hypopituitarism and QTc prolongation was investigated using univariate and multivariate logistic regression models. All clinical, biochemical, and hormonal variables were evaluated in the univariate analysis as potential predictors of prolonged QTc. In particular, the variable “hypopituitarism” was first analyzed as a binary variable (presence/absence), then according to disease severity—defined either by the absolute number of deficient hormonal axes or by the presence/absence of panhypopituitarism (defined as ≥4 deficiencies)—and finally by assessing individual hormonal deficiencies.

In the latter analysis, patients with a specific hormonal deficiency who were receiving adequate replacement therapy were classified as comparable to those without hormonal deficiency. In patients without formal evaluation of the somatotropic axis by stimulation testing, the presence of at least three other pituitary deficiencies was considered sufficient for the diagnosis of GHD, even in the absence of a stimulation test [61,62,63,64].

All variables with a *p*-value < 0.25 in the univariate analysis were entered into the multivariate models. In addition, and in accordance with the primary aim of the study, the different definitions of hypopituitarism, as described above, were included regardless of their univariate significance. In the multivariate analysis, a stepwise backward elimination algorithm was applied to remove variables that did not show an independent association with the outcome (*p*-value > 0.20).

Statistical analyses were performed using MedCalc^®^ Statistical Software version 20.007 (MedCalc Software Ltd., Ostend, Belgium).

## 3. Results

Between 1 April 2023, and 30 September 2024, at the Division of Endocrinology, Diabetes, and Metabolism, “Città della Salute e della Scienza” Hospital, Turin, a total of 209 ECG recordings were obtained from patients with a history of hypothalamic–pituitary disorders. Of these, 8 recordings were excluded due to a family history of sudden cardiac death or a personal history of cardiovascular events, and 16 were excluded because the patients were receiving medications potentially associated with QTc prolongation. Consequently, ECG tracings from 185 patients were analyzed (121 males; median age (IQR) 57.6 (23.4) years). Among these patients, 167 (90.3%) had some degree of hypopituitarism, whereas the remaining 18 (9.7%), although affected by pituitary disease, exhibited no hormonal deficiencies.

Figure 1 shows the CONSORT diagram of the study population.

### 3.1. Characteristics of the Study Population

The characteristics of the study population are summarized in Table 1.

No significant differences were observed between patients with and without hypopituitarism, except for a higher prevalence of males in the hypopituitarism group (*p* = 0.04). A higher prevalence of craniopharyngiomas, Rathke’s cleft cysts, and congenital central nervous system (CNS) malformations was also observed in patients with hypopituitarism; however, the overall distribution of pituitary pathologies prompting clinical evaluation did not reach statistical significance, although it approached the threshold (*p* = 0.07).

Prolonged QTc intervals (≥450 ms in males and ≥460 ms in females) were more frequent in the hypopituitarism group compared with patients without hypopituitarism (8.4% vs. 5.6%), although this difference was not statistically significant (*p* = 1.0).

A summary of the comparison between patients with and without hypopituitarism is presented in Table 1.

### 3.2. Characteristics of Patients with Prolonged QTc

No gender differences were observed between patients with prolonged QTc and those without (male 73.3% vs. 64.7%, *p* = 0.58). Patients with prolonged QTc were significantly older than those without (65.7 ± 24.2 vs. 57.0 ± 24.2 years, *p* = 0.04).

Among the biochemical parameters, patients with prolonged QTc had significantly lower total calcium (2.30 ± 0.13 vs. 2.39 ± 0.10 mmol/L, *p* = 0.02) and albumin-corrected calcium (2.27 ± 0.14 vs. 2.38 ± 0.11 mmol/L, *p* = 0.005) compared with patients without prolonged QTc. Potassium levels were also lower in the prolonged QTc group (3.9 ± 0.4 vs. 4.1 ± 0.4 mmol/L), although this difference did not reach statistical significance (*p* = 0.08). No significant differences were observed between groups in serum sodium, fT3, fT4, IGF-I, IGF-I SDS, or testosterone.

A history of neurosurgery and/or radiotherapy was more frequent in patients with prolonged QTc (73.3% vs. 49.4%), although this difference did not reach statistical significance (*p* = 0.10). The distribution of underlying pituitary pathologies potentially causing hypopituitarism did not differ significantly between groups (*p* = 0.47).

Similarly, the prevalence of pituitary hormone deficiencies did not differ significantly between the two groups, whether expressed as hypopituitarism (*p* = 1.0), panhypopituitarism (*p* = 0.60), or the number of pituitary deficits (*p* = 0.45). The prevalence of individual pituitary deficiencies and their respective replacement therapies also did not differ significantly (adrenal insufficiency, *p* = 0.08; hypothyroidism, *p* = 0.87; hypogonadism, *p* = 0.79; GHD, *p* = 0.73; diabetes insipidus, *p* = 0.44).

The results of the univariate analysis comparing patients with prolonged QTc to those with normal QTc are summarized in Table 2.

### 3.3. Correlations Between QTc Duration and Biochemical Parameters

In the overall study population, QTc showed a significant positive correlation with age (rho = 0.283, *p* = 0.0001; n = 185) and significant inverse correlations with potassium (r = −0.240, *p* = 0.017; n = 168), total calcium (r = −0.284, *p* = 0.005; n = 99), albumin-corrected calcium (r = −0.273, *p* = 0.01; n = 86), and absolute IGF-I levels (rho = −0.298, *p* = 0.005; n = 87). No statistically significant correlations were observed between QTc and sodium (rho = 0.003, *p* = 0.97; n = 171), fT3 (r = −0.021, *p* = 0.86; n = 76), fT4 (rho = 0.03, *p* = 0.78; n = 93), or IGF-I SDS (rho = −0.052, *p* = 0.63; n = 87).

In male subjects, no significant correlation was found between QTc duration and testosterone levels (r = 0.016, *p* = 0.91; n = 60).

### 3.4. Univariate Analysis of Predictors of Prolonged QTc

The results of the univariate analysis of predictors of prolonged QTc are summarized in Table 3. Age (OR = 1.05, 95% confidence interval (CI) 1.01–1.09, *p* = 0.03), a history of neurosurgery and/or radiotherapy (OR = 2.82, 95% CI 0.86–9.19), and IGF-I SDS (OR = 2.70, 95% CI 0.55–13.19) were identified as potential positive predictors of QTc prolongation, although the latter two associations only showed a trend toward statistical significance (*p* = 0.08 and 0.06, respectively).

Conversely, calcium levels, both when expressed as total calcium (OR = 0.0006, 95% CI 0.0–0.41, *p* = 0.03) and as albumin-corrected calcium (OR = 0.0002, 95% CI 0.0–0.14, *p* = 0.01), were associated with a lower likelihood of prolonged QTc. Potassium levels were also negatively associated with QTc prolongation (OR = 0.26, 95% CI 0.06–1.21), although this association approached statistical significance (*p* = 0.08).

The presence of pituitary hormone deficiencies was not significantly associated with QTc prolongation. However, a trend toward statistical significance (*p* = 0.08) was observed for untreated secondary adrenal insufficiency, which appeared as a potential positive predictor (OR = 12.07, 95% CI 0.72–203.5).

### 3.5. Multivariate Analysis of Predictors of Prolonged QTc

In the first three multivariate models using total calcium, age (OR 1.06, 95% CI 0.99–1.12, *p* = 0.08) and potassium levels (OR 0.15, 95% CI 0.02–1.31, *p* = 0.08) were independent, albeit borderline, predictors of prolonged QTc (Table 3). When pituitary deficiencies were analyzed as individual hormonal deficits, age (OR 1.07, 95% CI 1.01–1.14, *p* = 0.02) and the presence of expansive lesions other than pituitary adenomas, craniopharyngiomas, and Rathke’s cleft cysts (OR 8.35 vs. pituitary adenoma, 95% CI 1.0–70.13, *p* < 0.05) were significant predictors, while potassium remained a borderline predictor (OR 0.17, 95% CI 0.02–1.36, *p* = 0.09) (Table 3).

In multivariate models including albumin-corrected calcium, regardless of how pituitary deficiencies were defined (presence/absence of hypopituitarism, panhypopituitarism, number of hormonal deficits, or specific hormonal deficits), age (OR 1.09, 95% CI 1.01–1.18, *p* = 0.02) and the presence of expansive lesions other than pituitary adenomas, craniopharyngiomas, and Rathke’s cleft cysts (OR 17.73 vs. pituitary adenoma, 95% CI 1.32–238.54, *p* = 0.03) emerged as independent predictors of prolonged QTc. Potassium (OR 0.14, 95% CI 0.02–1.44, *p* = 0.09) and albumin-corrected calcium levels (OR 0.0003, 95% CI 0.0–1.20, *p* = 0.06) showed borderline associations (Table 3).

## 4. Discussion

Our findings suggest that, in patients with a history of hypothalamic–pituitary disease, QTc prolongation is associated with clinical variables that are well established in the general population, such as age. Consistent with this, in our cohort age correlated positively with QTc duration (ρ = 0.283, *p* = 0.0001) and remained an independent predictor in multivariate analyses (OR 1.09, 95% CI 1.01–1.18, *p* = 0.02). Interestingly, certain underlying hypothalamic–pituitary pathologies—particularly expansive lesions other than pituitary adenomas, craniopharyngiomas, and Rathke’s cleft cysts—were also independently associated with QTc prolongation after adjustment for age (OR range 8.35–17.73, *p* < 0.05 in multivariate models), suggesting that structural hypothalamic or parasellar damage may directly influence cardiac repolarization. Moreover, lower serum potassium (OR range 0.14–0.17, *p* = 0.08–0.09) and reduced albumin-corrected calcium (OR 0.0003, *p* = 0.06) showed borderline inverse associations with QTc prolongation, indicating that even mild electrolyte imbalances may modulate QTc duration independently of age. Nevertheless, the impact of pituitary hormone deficiencies on QTc prolongation remains uncertain, as most affected patients were already receiving appropriate hormone replacement therapy at the time of ECG acquisition.

To the best of our knowledge, this is the first study to systematically evaluate potential predictors of QTc interval prolongation in patients with hypopituitarism. The currently available literature linking hypopituitarism to aLQTS is limited to 16 case reports [42,43,44,45,46,47,48,49,50,51,52,53,54,55,56,57], which are highly heterogeneous in terms of hormonal deficiencies, diagnostic criteria, and the presence of potential confounding factors, such as comorbidities and concomitant medications. To minimize the impact of external confounders, we excluded patients receiving medications known to potentially prolong the QT interval, as well as those with a family history of sudden cardiac death or a personal history of cardiovascular events. Furthermore, since the primary aim of our study was to specifically assess the impact of hypopituitarism on QTc prolongation, we excluded patients with pituitary disorders associated with active hormone hypersecretion and hyperthyroidism, as these conditions themselves influence QT interval duration [26,27,28,29,30,31,32,33,34,35,36,37,38,39,40,41].

Our results confirm previously established associations in the general population between QTc duration and clinical parameters such as age [17,24,25]. However, in our study, age did not fully account for QTc variability, as additional factors—namely hypothalamic–parasellar structural lesions and subtle electrolyte alterations—also contributed to QTc prolongation after age adjustment. Electrolyte levels, particularly potassium and albumin-corrected calcium [17,24,25], appeared to contribute to QTc duration to a lesser extent in patients with hypothalamic–pituitary disorders. It should be noted, however, that in our cohort the prevalence of hypokalemia (potassium ≤ 3.5 mmol/L) and hypocalcemia (albumin-corrected calcium ≤ 2.20 mmol/L) was relatively low (4.8% and 7%, respectively), and no severe cases (potassium ≤ 3.0 mmol/L; albumin-corrected calcium ≤ 2.0 mmol/L) were observed.

Furthermore, we identified disease-specific factors in patients with hypothalamic–pituitary disorders, including certain underlying pathologies such as expansive lesions other than pituitary adenomas, craniopharyngiomas, and Rathke’s cleft cysts. This association remained significant after controlling for age, suggesting a pathophysiological link possibly mediated by hypothalamic injury or altered autonomic regulation.

Although the prevalence of QTc prolongation appeared higher than that reported in the general population [65], no statistically significant difference was detected between patients with and without hypopituitarism. The precise role of pituitary hormone deficiencies remains to be clarified, as most affected patients were already receiving appropriate replacement therapy at the time of ECG assessment.

Taken together, our findings indicate that, while age is the strongest determinant of QTc prolongation, additional factors such as structural hypothalamic involvement and mild electrolyte disturbances may further contribute to QTc variability, likely through combined effects on autonomic tone and ventricular repolarization.

Sex is known to influence QT interval duration, with women generally exhibiting slightly longer QTc intervals [17,59], likely due to hormonal factors, particularly the effects of estrogens. In our cohort, however, gender distribution did not differ significantly between patients with and without QTc prolongation. Although univariate analysis revealed a direct correlation between age and QTc duration, no significant age difference was observed between men and women, suggesting that the lack of gender-related differences in QTc prolongation was independent of age. Furthermore, in most multivariate models, age consistently emerged as an independent predictor of QTc prolongation, whereas sex did not.

The absence of gender differences between patients with and without prolonged QTc may be partly explained by differences in the management of hypogonadism. Although the overall prevalence of treated versus untreated hypogonadism did not differ between patient groups—and this remained true after stratification by sex—the proportion of treated hypogonadism was substantially higher in men (77.5%) than in women (27.8%). As estrogen levels are a major determinant of sex-related differences in QT duration [17,59], with women typically exhibiting longer QTc intervals than men, the lack of gonadal replacement therapy in females may have contributed to the similar gender distribution observed in the two groups.

Another expected and well-documented association is that between electrolyte disturbances and QTc duration. Patients with lower potassium, calcium, and/or albumin-corrected calcium levels tended to have longer QTc intervals, consistent with evidence from the general population showing that hypokalemia, hypocalcemia, and hypomagnesemia [17,66] are established risk factors for arrhythmias. Unfortunately, magnesium levels were not available in our cohort, as this parameter is not routinely monitored in hypopituitary patients. In both univariate and multivariate analyses, potassium and albumin-corrected calcium levels were inversely associated with QTc duration, although the associations occasionally reached only borderline statistical significance. Total calcium levels were significantly lower in patients with prolonged QTc in univariate analysis but lost significance in multivariate models.

Regarding hormonal factors, IGF-I was the only parameter showing an inverse correlation with QTc duration, suggesting a potential link between lower IGF-I levels and longer QTc intervals. However, this association is most likely explained by the physiological decline of IGF-I with aging [60,67], rather than by a direct effect of GHD. Indeed, once IGF-I values were adjusted for age (expressed as SDS), the association with QTc disappeared, indicating that age, rather than IGF-I per se, is the main determinant of this relationship. Notably, IGF-I SDS levels were significant predictors of prolonged QTc in univariate analysis but lost statistical significance across the various multivariate models examined.

No significant associations were observed between QTc duration and other biochemical markers, including sodium, thyroid hormones, and testosterone, suggesting that these parameters do not directly influence QTc in this population. However, considering that almost all patients (97.9% of those with adrenal insufficiency, 97.1% with hypothyroidism, 77.5% of men with hypogonadism, and 100% of those with diabetes insipidus) were receiving adequate hormone replacement therapy, the absence of differences between patients with and without hypopituitarism is not unexpected, as replacement therapy likely normalized these biochemical variables.

In our cohort, the prevalence of hypopituitarism was high (90.1%), confirming that nearly all patients had underlying conditions predisposing them to pituitary hormone deficiencies. Although the prevalence of prolonged QTc appeared higher than expected [65], no significant differences were observed between patients with and without hypopituitarism. This finding suggests that hypopituitarism per se is not a determinant of QTc duration, a conclusion further supported by multivariate analyses, which showed no association between hypopituitarism and QTc prolongation. Similarly, neither panhypopituitarism nor the number of deficient hormonal axes correlated with prolonged QTc, indicating that QTc duration is not influenced by the overall severity of hormonal deficiencies. Nevertheless, a potential association between individual pituitary hormone deficiencies—particularly hypocortisolism, hypothyroidism, and diabetes insipidus—and QTc prolongation in patients with hypothalamic–pituitary disease cannot be entirely ruled out, as almost all patients with these conditions were already receiving hormone replacement therapy at the time of ECG recording. Regarding hypogonadism, the absence of an association in our cohort—characterized by a high prevalence of replacement therapy, particularly among men—indirectly supports previous evidence indicating that testosterone replacement mitigates the adverse cardiac effects of hypogonadism [68] and has been reported to correct QTc prolongation in hypogonadal men [58]. As for GHD, it is noteworthy that, unlike other pituitary hormone deficiencies, a substantial proportion of patients remained untreated (55.7% of GHD cases). The lack of any observed association between GHD and QTc prolongation may therefore suggest that GHD, even when untreated, does not significantly influence QT interval duration in patients with hypothalamic–pituitary disorders. This interpretation is further supported by published case reports describing normalization of the QTc interval even in the absence of GH replacement therapy [42,43,44,45,47,50,54].

Finally, in at least five of the eight proposed models, the presence of expansive lesions other than pituitary adenomas, craniopharyngiomas, or Rathke’s cleft cysts emerged as an independent predictor of prolonged QTc. This finding does not appear to be attributable to a history of more invasive treatments, which could have caused direct or indirect damage to the neurological circuits involved in cardiovascular regulation. Instead, it may suggest a potential intrinsic effect of these alternative expansive lesions on cardiac repolarization. Nevertheless, this association should be interpreted with caution and warrants confirmation in larger, independent cohorts.

The main strength of this study is the relatively large sample size for a rare condition such as hypopituitarism. Moreover, the single-center design over a relatively short period allowed for a consistent comparison of biochemical and instrumental parameters. Limitations include its retrospective, observational nature, which resulted in incomplete biochemical data for some patients at the time of ECG acquisition. Additionally, as previously noted, the high proportion of hypopituitary patients already receiving hormone replacement therapy at the time of ECG recording may have masked potential correlations between individual pituitary hormone deficiencies and QTc prolongation. Given the cross-sectional and retrospective design of the study, our available data do not allow evaluation of potential improvements in QTc parameters during hormone replacement therapy. To address whether replacement therapy can normalize any QTc prolongation in hypopituitary patients, further longitudinal studies are required.

## 5. Conclusions

This study confirms that age and electrolyte disturbances influence QTc duration in patients with a history of hypothalamic–pituitary disease, consistent with observations in the general population. Notably, the presence of expansive lesions other than pituitary adenomas, craniopharyngiomas, and Rathke’s cleft cysts emerged as an independent predictor of prolonged QTc, although these findings require confirmation in larger, prospective studies. Despite the apparently higher prevalence of QTc prolongation compared with the general population, no significant differences were observed between patients with and without hypopituitarism. The role of pituitary hormone deficiencies remains unclear, as most affected patients were already receiving replacement therapy at the time of ECG acquisition. However, this study emphasizes the increased arrhythmogenic risk in patients with hypopituitarism.

These findings highlight the importance of careful monitoring and appropriate management of electrolyte imbalances and hormonal deficiencies to prevent QTc prolongation and reduce the risk of potentially life-threatening arrhythmias in this population. Importantly, some commonly used diagnostic tests for hypopituitarism—such as the insulin tolerance test and the macimorelin test—have been reported to potentially prolong the QTc interval [69,70]. Therefore, identifying additional predictors of QTc prolongation in hypopituitary patients is essential to develop a diagnostic pathway that is broadly applicable, safe, and tailored to the specific needs of each patient.

## Figures and Tables

**Figure 1 biomedicines-13-02676-f001:**
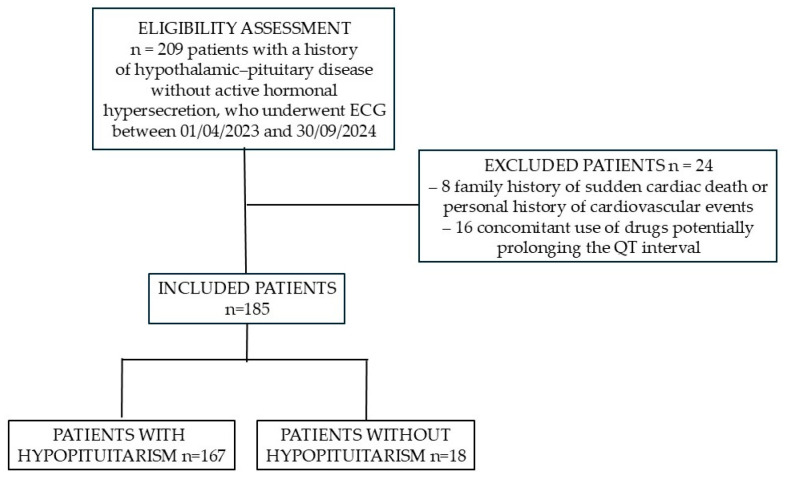
CONSORT flow diagram of the study population.

**Table 1 biomedicines-13-02676-t001:** Clinical characteristics of the study population, comparing patients with and without hypopituitarism.

	All (n = 185)	Patients with Hypopituitarism (n = 167)	Patients Without Hypopituitarism (n = 18)	*p*
Male gender, n (%)	121 (65.4)	105 (62.9)	16 (88.9)	0.04
Age (yrs)	57.6 (23.4)	57.0 (25.5)	59.5 (14.5)	0.20
Heart rate (pulse/min)	69.6 ± 12.3	69.9 ± 12.4	66.7 ± 10.5	0.28
Bazett QTc (ms)	416.9 ± 27.8	416.9 ± 27.4	416.3 ± 32.3	0.92
Sodium (mmol/L)	140.0 (4.0) (n = 171)	140.0 (4.0) (n = 155)	140.5 (4.0) (n = 16)	0.69
Potassium (mmol/L)	4.1 ± 0.4 (n = 168)	4.1 ± 0.4 (n = 152)	4.0 ± 0.4 (n = 16)	0.83
Calcium (mmol/L)	2.38 ± 0.11 (n = 99)	2.38 ± 0.11 (n = 95)	2.37 ± 0.11 (n = 4)	0.88
Albumin-corrected calcium (mmol/L)	2.37 ± 0.12 (n = 86)	2.37 ± 0.12 (n = 83)	2.32 ± 0.02 (n = 3)	0.48
Free T3 (ng/L)	2.7 ± 0.6 (n = 76)	2.7 ± 0.6 (n = 76)	No data available	/
Free T4 (ng/L)	9.1 (3.0) (n = 93)	9.2 (3.0) (n = 91)	8.5 (1.0) (n = 2)	0.29
IGF-I (µg/L)	148.9 (128.1) (n = 87)	145.2 (119.9) (n = 84)	219.9 (152.9) (n = 3)	0.35
IGF-I SDS	0.02 (1.27) (n = 87)	0.005 (1.27) (n = 84)	0.58 (0.69) (n = 3)	0.16
Testosterone (µg/L)	4.3 ± 2.8 (n = 60)	4.2 ± 2.8 (n = 57)	5.0 ± 2.8 (n = 3)	0.62
Neurosurgery and/or Radiotherapy, n (%)	95 (51.4)	89 (52.7)	6 (33.3)	0.14
Pituitary disease, n (%)				0.07
-Pituitary adenoma	114 (61.6)	97 (58.1)	17 (94.4)	
-Craniopharyngioma or Rathke’s cleft cyst	15 (8.1)	15 (9.0)	0 (0.0)	
-Other types of expansive lesions *	10 (5.4)	9 (5.4)	1 (5.6)	
-CNS malformation	40 (21.6)	40 (23.9)	0 (0.0)	
-Idiopathic isolated GHD	2 (1.1)	2 (1.2)	0 (0.0)	
-Miscellaneous	4 (2.2)	4 (2.4)	0 (0.0)	
Panhypopituitarism, n (%)	85 (46.0)	85 (50.9)	0 (0.0)	<0.0001
Number of pituitary deficit, n (%)				<0.0001
-0	18 (9.7)	0 (0.0)	18 (100.0)	
-1	39 (21.1)	39 (23.3)	0 (0.0)	
-2	32 (17.3)	32 (19.2)	0 (0.0)	
-3	11 (5.9)	11 (6.6)	0 (0.0)	
-4	68 (36.8)	68 (40.7)	0 (0.0)	
-5	17 (9.2)	17 (10.2)	0 (0.0)	
Type of pituitary deficit, n (%)				
-Hypocortisolism				<0.0001
*not on therapy*	2 (1.1)	2 (1.2)	0 (0.0)	
*on therapy*	92 (49.7)	92 (55.1)	0 (0.0)	
-Hypothyroidism				<0.0001
*not on therapy*	3 (1.6)	3 (1.8)	0 (0.0)	
*on therapy*	99 (53.5)	99 (59.3)	0 (0.0)	
-Hypogonadism				<0.0001
*not on therapy*	57 (30.8)	57 (34.1)	0 (0.0)	
*on therapy*	77 (41.6)	77 (46.1)	0 (0.0)	
-GHD				<0.0001
*not on therapy*	68 (36.8)	68 (40.7)	0 (0.0)	
*on therapy*	54 (29.2)	54 (32.3)	0 (0.0)	
*not assessed*	52 (28.1)	36 (21.6)	16 (88.9)	
-Diabetes insipidus **	26 (14.1)	26 (15.6)	0 (0.0)	0.08
QTc ≥ 450 ms in male and ≥460 ms in female, n (%)	15 (8.1)	14 (8.4)	1 (5.6)	1.0

* Other types of expansive lesions: 2 germinomas, 2 pituitary mass lesions not further characterized, 1 tuberculum sellae meningioma, 1 granular cell tumor of the infundibulum, 1 sphenoid chondrosarcoma, 1 case with sequelae of fourth ventricle ependymoma treatment, 1 thickening of the pituitary stalk, and 1 epidermoid cyst. ** All cases of diabetes insipidus were treated; CNS: central nervous system; GHD: growth hormone deficiency; IGF-I: insulin like growth factor-I; ms: milliseconds; SDS: standard deviation score; T3: triiodothyronine; T4: thyroxine; yrs: years.

**Table 2 biomedicines-13-02676-t002:** Distribution of the various characteristics analyzed in patients with prolonged QTc (≥450 ms in males, ≥460 ms in females) compared with those with normal QTc values.

	Prolonged QTc (n = 15)	Normal QTc (n = 170)	*p*
Male gender, n (%)	11 (73.3)	110 (64.7)	0.58
Age (yrs)	65.7 (24.2)	57.0 (24.2)	0.04
Heart rate (pulse/min)	73.7 ± 9.7	69.3 ± 12.4	0.17
Bazett QTc (ms)	472.1 ± 17.5	412.0 ± 22.8	<0.0001
Sodium (mmol/L)	140.0 (2.8) (n = 15)	140.0 (4.0) (n = 156)	0.72
Potassium (mmol/L)	3.9 ± 0.4 (n = 14)	4.1 ± 0.4 (n = 154)	0.08
Calcium (mmol/L)	2.30 ± 0.13 (n = 9)	2.39 ± 0.10 (n = 90)	0.02
Albumin-corrected calcium (mmol/L)	2.27 ± 0.14 (n = 9)	2.38 ± 0.11 (n = 77)	0.02
Free T3 (ng/L)	2.5 ± 0.6 (n = 7)	2.7 ± 0.5 (n = 69)	0.38
Free T4 (ng/L)	9.5 (2.3) (n = 8)	9.1 (3.0) (n = 85)	0.69
IGF-I (µg/L)	120.0 (65.3) (n = 7)	152.4 (133.5) (n = 80)	0.63
IGF-I SDS	0.31 (0.34) (n = 7)	−0.06 (1.29) (n = 80)	0.17
Testosterone (µg/L)	5.2 ± 3.1 (n = 6)	4.1 ± 2.7 (n = 54)	0.37
Neurosurgery and/or Radiotherapy, n (%)	11 (73.3)	84 (49.4)	0.10
Pituitary disease, n (%)			0.47
-Pituitary adenoma	11 (73.3)	103 (60.6)	
-Craniopharyngioma or Rathke’s cleft cyst	1 (6.7)	14 (8.2)	
-Other types of expansive lesions *	2 (13.3)	8 (4.7)	
-CNS malformation	1 (6.7)	39 (22.9)	
-Idiopathic isolated GHD	0 (0.0)	2 (1.2)	
-Miscellaneous	0 (0.0)	4 (2.4)	
Hypopituitarism, n (%)	14 (93.3)	153 (90.0)	1.0
Panhypopituitarism, n (%)	8 (53.3)	77 (45.3)	0.60
Number of pituitary deficit, n (%)			0.45
-0	1 (6.7)	17 (10.0)	
-1	5 (33.3)	34 (20.0)	
-2	0 (0.0)	32 (18.8)	
-3	1 (6.7)	10 (5.9)	
-4	7 (46.6)	61 (35.9)	
-5	1 (6.7)	16 (9.4)	
Type of pituitary deficit, n (%)			
-Hypocortisolism			0.08
*not on therapy*	1 (6.7)	1 (0.6)	
*on therapy*	6 (40.0)	86 (50.6)	
-Hypothyroidism			0.79
*not on therapy*	0 (0.0)	3 (1.8)	
*on therapy*	8 (53.3)	91 (53.2)	
-Hypogonadism			0.79
*not on therapy*	5 (33.3)	52 (30.6)	
*on therapy*	7 (46.7)	70 (41.2)	
-GHD			0.73
*not on therapy*	5 (33.3)	63 (37.1)	
*on therapy*	5 (33.3)	49 (28.8)	
*not* assessed	5 (33.3)	47 (27.6)	
-Diabetes insipidus **	3 (20.0)	23 (13.5)	0.44

* Other types of expansive lesions: 2 germinomas, 2 pituitary mass lesions not further characterized, 1 tuberculum sellae meningioma, 1 granular cell tumor of the infundibulum, 1 sphenoid chondrosarcoma, 1 case with sequelae of fourth ventricle ependymoma treatment, 1 thickening of the pituitary stalk, and 1 epidermoid cyst. ** All cases of diabetes insipidus were treated; CNS: central nervous system; GHD: growth hormone deficiency; IGF-I: insulin like growth factor-I; ms: milliseconds; SDS: standard deviation score; T3: triiodothyronine; T4: thyroxine; yrs: years.

**Table 3 biomedicines-13-02676-t003:** Univariate and multivariate logistic regression for predictors of prolonged QTc (≥450 ms in males, ≥460 ms in females). All variables with a *p*-value < 0.25 in the univariate analysis and the different definitions of hypopituitarism were entered into the multivariate models (stepwise backward elimination algorithm) regardless of their univariate significance.

LONG QTc	Univariate Analysis		Multivariate Model # 1, 2, 3		Multivariate Model # 4		Multivariate Model # 5, 6, 7, 8	
	OR	*p* Value	OR	*p* Value	OR	*p* Value	OR	*p* Value
Age	1.05	0.03	1.06	0.08	1.07	0.02	1.09	0.02
Male sex	1.5	0.50	-	-	-	-	-	-
Sodium levels	0.94	0.56	-	-	-	-	-	-
Potassium levels	0.26	0.08	0.15	0.08	0.17	0.09	0.14	0.09
Calcium levels	0.0006	0.03	-	ns	-	ns	NA	NA
Albumin corrected calcium levels	0.0002	0.01	NA	NA	NA	NA	0.0003	0.06
Free T3	0.45	0.34	-	-	-	-	-	-
Free T4	0.98	0.93	-	-	-	-	-	-
IGF-I	0.99	0.39	-	-	-	-	-	-
IGF-I SDS	2.70	0.06	-	ns	-	ns	-	ns
Testosterone	1.13	0.39	-	-	-	-	-	-
Neurosurgery and/or Radiotherapy	2.82	0.08	3.31	0.19	-	ns	-	ns
Pituitary disease (vs. pituitary adenoma)								
-Craniopharyngioma or Rathke’s cleft cyst	0.67	0.71	-	ns	-	ns	-	ns
-Other type of expansive lesions *	2.34	0.32	5.11	0.15	8.35	<0.05	17.73	0.03
-CNS malformation	0.24	0.18	-	ns	-	ns	-	ns
-Idiopathic isolated GHD	0.0	0.99	-	ns	-	ns	-	ns
-Miscellaneous	0.0	0.99	-	ns	-	ns	-	ns
Hypopituitarism	1.56	0.68	-	ns	NA	NA	-	ns
Panhypopituitarism	1.38	0.55	-	ns	NA	NA	-	ns
N. of pituitary deficit (vs. 0)					NA	NA		
-1	2.5	0.42	-	ns			-	ns
-2	0	0.99	-	ns			-	ns
-3	1.7	0.72	-	ns			-	ns
-4	1.95	0.54	-	ns			-	ns
-5	1.06	0.97	-	ns			-	ns
Type of pituitary deficit (no vs. yes)								
-Hypocortisolism	12.07	0.08	NA	NA	-	-	-	ns
-Hypothyroidism	0	0.99	NA	NA	-	-	-	ns
-Hypogonadism	1.13	0.83	NA	NA	-	-	-	ns
-GHD								
*no* vs. *yes*	0.95	0.94	NA	NA	-	-	-	ns
*no* vs. *not assessed*	1.28	0.71	NA	NA	-	-	-	ns
-Diabetes insipidus	1.60	0.49	NA	NA	-	-	-	ns

Variables in multivariate analysis: age (model # 1–8), potassium (model # 1–8), total calcium (model # 1–4), albumin-corrected calcium (model # 5–6), prior neurosurgery/radiotherapy (model # 1–8), underlying pituitary disease (model # 1–8), and pituitary hormone deficiencies, categorized as hypopituitarism (present/absent) (model # 1, 5), panhypopituitarism (present/absent) (model # 2, 6), number of deficient axes (model # 3, 7), or specific deficiencies (adrenal insufficiency, hypothyroidism, hypogonadism, GHD, diabetes insipidus) (model # 4, 8); Multivariate model # 1, 2, 3: N. of observations = 91, LR chi2 (4) = 13.208, *p* = 0.01; Multivariate model # 4: N. of observations = 91, LR chi2 (3) = 11.533, *p* = 0.009; Multivariate model # 5, 6, 7, 8: N. of observations = 81, LR chi2 (4) = 17.023, *p* = 0.0019; * Other types of expansive lesions: 2 germinomas, 2 pituitary mass lesions not further characterized, 1 tuberculum sellae meningioma, 1 granular cell tumor of the infundibulum, 1 sphenoid chondrosarcoma, 1 case with sequelae of fourth ventricle ependymoma treatment, 1 thickening of the pituitary stalk, and 1 epidermoid cyst. CNS: central nervous system; GHD: growth hormone deficiency; LR: likelihood ratio; N.: number; NA: not applicable; ns: not significant; OR: odds ratio.

## Data Availability

The data sets generated and analyzed during the current study are not publicly available but are available from the corresponding author on reasonable request.

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
