# Peer review of "Determinants of QTc Interval Prolongation in Patients with Hypopituitarism and Other Pituitary Disorders"

_biomedicines, 2025, doi:10.3390/biomedicines13112676_

Round 1

Reviewer 1 Report

Comments and Suggestions for Authors

The authors analyze the link between hypothalamic–pituitary disorders and corrected QT (QTc) interval. The paper is original, English very correct, the text is written in understandable manner. Furthermore, the article is significant for clinical practice since it emphasizes not just coronary increased cardiovascular risk, but also arrhythmogenic in hypopituitarism patients.

Two minor issues: 

  1. show us prolactin levels and its impact on cQTc
  2. spell out SNC in Table 2. 

Reviewer 2 Report

Comments and Suggestions for Authors

The authors are investigating a known marker predisposing to malignant arrhythmias, long QT syndrome (LQTS) in patients with Hypothalamic-Pituitary Disorders. However, the title and the text are not clearly presenting the topic of the research since the reader cannot know if the disorders are relating to hormonal hyposecretion or hypersecretion. The confusion increases when the introduction defines only ‘Hypopituitarism’. Please correct the title and the misspelling in the title I believe that will be deleted.

In addition, the authors are stating that there is no clear cut-off to define an abnormal QT interval in men and women so the authors may have to think to include a group of healthy subjects and compare the QT and they will avoid the comparison on a group of 167 people with 18 people.

Another point not clear is the definition of ‘expansive lesions other than pituitary adenomas, craniopharyngiomas, and Rathke’s cleft cysts’. This negative definition is confusing as opposed to the need that the reader has to understand any differences in the context of LQTS in the adenomas versus parasellar lesions. Finally, the disorders described here have origin from adenomas that for some of them have been described, so the authors have to proceed to a revision of the literature to provide data on hyperfunction and the outcome post-surgery. For example, the LQTS has been describe in pituitary disorders in long series as in 79 patients with Cushing’s disease (DOI: 10.1111/j.1365-2265.2011.03975.x). Please explain if you may speculate from this or others studies whether the syndrome preexists and it is not due to the pituitary insufficiency, if it is remitted when the disorder is cured without pituitary insufficiency or if it is because of overtreatment as it could be the results of replacement post-adrenal insufficiency. Please discuss all the possibilities with the evidence of your study.

Additional minor comments

Methods

‘Patients receiving medications known to potentially prolong the QT interval’ do you use a particular list in the protocol of the study?

Results

In previous studies no differences were seen in the context of electrolytes disturbances; can you please study your subgroups to give more clear data?

Please present in the text the parameters used in the univariate binary regression analysis and in the test only the significant and the trends to statistically significant differences. In the multivariate analysis use only the statistically significant differences. Please merge uni-and multivariate analysis in the same table.

Comments on the Quality of English Language

check some mispelling

Reviewer 3 Report

Comments and Suggestions for Authors

The authors reviewed the clinical characteristics (including corrected QT interval) in patients with hypothalamic-pituitary disorders. However, the clinical relevance of the findings and the potential implications are not clearly shown. Additionally, the discussion section should be rephrased to improve clarity.  

Round 2

Reviewer 2 Report

Comments and Suggestions for Authors

The manuscript is much improved but some minor comments are necessary to improve the quality of the manuscript. In the way the authors have corrected the manuscript and since they still include the 10% of the patients without hypopituitarism the title has to be amended as ‘Determinants of QTc Interval Prolongation in Patients with Hypopituitarism and other pituitary disorders’ or exclude the minority without hypopituitarism and keep the current title.

More minor comments

Lines from the first paragraph of page 13: (such as Cushing’s dis-371 ease, acromegaly, hyperprolactinemia, and hyperthyroidism) since you reported that on Methods section

Author Response

We would like to sincerely thank the Reviewer for the careful reading of our manuscript and for the helpful comments that have contributed to improving its quality.

Comment 1:
The manuscript is much improved but some minor comments are necessary to improve the quality of the manuscript. In the way the authors have corrected the manuscript and since they still include the 10% of the patients without hypopituitarism the title has to be amended as ‘Determinants of QTc Interval Prolongation in Patients with Hypopituitarism and other pituitary disorders’ or exclude the minority without hypopituitarism and keep the current title.

Response:
We thank the Reviewer for this valuable suggestion. In accordance with the Reviewer’s recommendation, we have modified the title of the manuscript to “Determinants of QTc Interval Prolongation in Patients with Hypopituitarism and Other Pituitary Disorders.”

Comment 2:
Lines from the first paragraph of page 13: (such as Cushing’s disease, acromegaly, hyperprolactinemia, and hyperthyroidism) since you reported that on Methods section.

Response:
We appreciate the Reviewer’s comment. Although we were not entirely certain about the exact intention of this suggestion, we interpreted it as an indication to remove the mentioned text from page 13, as these conditions had already been described in the Methods section. We have therefore deleted the corresponding part of the text. Nevertheless, we remain fully available to make further adjustments should we have misunderstood the Reviewer’s request.